# FootCapture: Towards an AR-based System for 3D Foot Object Acquisition through Photogrammetry

**Valentin Khan-Blouki**[*,1]                    VALENTIN.KHAN-BLOUKI@STUDENT.KIT.EDU
**Franziska Seiz**[*,1]                          FRANZISKA.SEIZ@STUDENT.KIT.EDU
**Nicolas Walter**[*,1]                          NICOLAS.WALTER@STUDENT.KIT.EDU
**Alexander Jaus**[1,2]                          ALEXANDER.JAUS@KIT.EDU
**Zdravko Marinov**[1,2]                         ZDRAVKO.MARINOV@KIT.EDU
**Gijs Luijten**[3]                              GIJS.LUIJTEN@UK-ESSEN.DE
**Jan Egger**[3]                                 JAN.EGGER@UK-ESSEN.DE
**Constantin Seibold**[3]                        CONSTANTIN.SEIBOLD@UK-ESSEN.DE
**Dirk Solte**[4]                                DIRK.SOLTE@KNOWING.DE
**Jens Kleesiek**[3]                             JENS.KLEESIEK@UK-ESSEN.DE
**Rainer Stiefelhagen**[1]                       RAINER.STIEFELHAGEN@KIT.EDU

[1] *Institute for Anthropomatics & Robotics (IAR), Karlsruhe Institute of Technology, Germany*

[2] *HIDSS4Health - Helmholtz Information and Data Science School for Health, Karlsruhe, Germany*

[3] *Institute for AI in Medicine (IKIM), University Hospital Essen, Germany*

[4] *Knowing GmbH, Karlsruhe, Germany*

## Abstract

The acquisition of accurate 3D models of feet is crucial in fields such as chronic foot wound monitoring, prosthetics design, and orthopedic surgery. However, obtaining precise models of patients' feet typically relies on manual measurements, which is both costly and prone to error. Addressing this need, we introduce *FootCapture*, a mobile application designed to facilitate the acquisition of precise photographic measurements. Our solution employs augmented reality to intuitively guide untrained users to capture comprehensive photographic data from the correct positions and angles, suitable to create a high-fidelity 3D model of the patient's foot using photogrammetry. To validate our application's utility, we compared *FootCapture* with Apple's *Guided Capture* application in a user study with $n = 7$ participants. The results showed FootCapture's intuitive use and high robustness marking it as a tool worth considering for medical personnel.

**Keywords:** Augmented reality, 3D object acquisition, foot monitoring, photogrammetry.

## 1. Introduction

The treatment and monitoring of chronic wounds pose significant challenges to both medical personnel and the healthcare system. It is estimated that only in the UK, the treatment of patients suffering from chronic wounds costs the NHS over seven billion GBP annually (Guest et al., 2020). The assessment of the wound healing process is a non-trivial task with traditional and rather subjective methods such as visual inspection (Ud-Din and Bayat, 2016) or simple scales (Pillen et al., 2009). Efforts to improve wound healing assessments

---

[*] Contributed equally

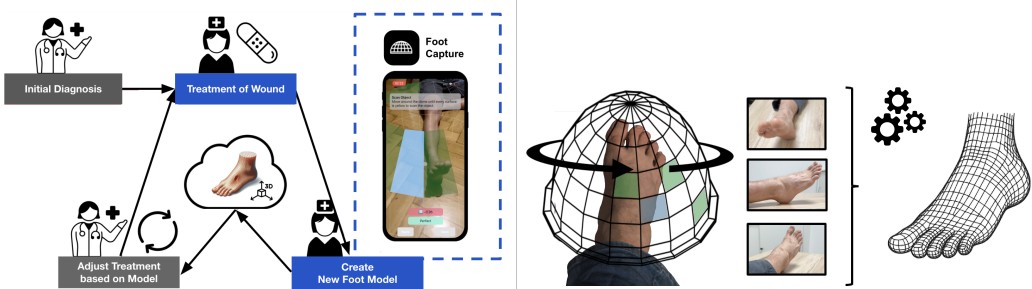

Figure 1: Our app, FootCapture, promotes a location-independent workflow for treating chronic wounds, monitoring the healing process, and adjusting their treatment.

have moved towards 3D imaging to capture the wound's intricate surface structure, addressing the limitations of basic 2D photographs. However, 3D imaging requires specialized equipment like lasers or a multitude of suitable images for 3D rendering, posing challenges for high-quality model creation (Treuillet et al., 2009; Tan et al., 2021; Kecelj-Leskovec et al., 2007). Our FootCapture app leverages recent smartphones' LiDAR technology to help caregivers take depth-based images that can produce high-quality 3D models of the foot and lower leg—critical areas for chronic wounds—without any prior training. Applications like this can help enable remote assessments and consistent monitoring of wound progression which would improve patient care (Etufugh and Phillips, 2007; Jeffcoate and Harding, 2003).

## 2. Method

When using the FootCapture app, comprehensive image and depth information is collected which then can be exported to another device (e.g. a laptop or server), to render a 3D model. As shown in figure 1, a *dome*, meaning a virtual, stretched hemisphere, is placed closely encapsulating the patient's foot. Each dome tile represents a target image to be captured. The user is guided by the tiles and textual instructions for the pitch and yaw angles to an ideal phone position in which the app automatically captures image and depth information. The respective tile turns green indicating progress and the next closest target to be captured is highlighted. Optionally a user can take additional pictures of particularly relevant foot areas. After acquisition, the image data has to be exported to be rendered, preferably using the Apple Object Capture API [1]. Implementation details can be found in the project's GitHub repository[2].

## 3. Evaluation Setup and Results

To evaluate the effectiveness and usability of our FootCapture app, we compared it to Apple's *Guided Capture*[3] app via a comparative user study which was conducted at the University Hospital Essen with seven participants, resulting in seven foot scans from each

---

1. https://developer.apple.com/augmented-reality/object-capture/
2. https://github.com/ValentinKhanBlouki/AR_footcapture
3. https://developer.apple.com/videos/play/wwdc2023/10191/

mobile app. We compare usability in terms of SEQ and NASA-TLX, and the quality of 3D foot models via Hausdorff distance (HD) and normalized surface distance (NSD) between foot models of the two apps and the Artec Leo scanner[4], a state of the art 3D Scanner.

### 3.1. User Study

Based on the participants' quantitative feedback, the SEQ scores were $5.0 \pm 1.0$ for Foot-Capture and $5.0 \pm 1.2$ for GuidedCapture, while NASA-TLX scores were $40.5 \pm 6.4$ for FootCapture and $39.7 \pm 13.9$ for GuidedCapture. Both FootCapture and GuidedCapture performed similarly well in these metrics, a positive outcome considering Apple's reputation for creating products users love. FootCapture was praised for its straightforward geometric concept but received criticism for its pitch and yaw angle feedback while Apple's UI design was liked, while the placement of its bounding box was deemed cumbersome.

### 3.2. Mesh Comparison

The meshes of the 3D models were quantitatively evaluated using MeshLab by registering them onto the ground truth created by the medical scanner. The results, presented in Table 1 indicate that the foot models created using FootCapture tend to be superior to the ones created by Apple's GuidedCapture app. Among the foot models generated by GuidedCapture, there is one particularly unsuccessful scan (M1), highlighting the robustness of our approach which is vital in a limited-time scenario which is typical in the field of patient care. However, even when omitting the GuidedCapture app's major failure M1, our approach still performs on average slightly better indicating the suitability of the FootCapture application.

|  | M1 | M2 | M3 | M4 | M5 | M6 | M7 | $\mathbf{M_{[1,7]}}$ | $\mathbf{M_{[2,7]}}$ |
|---|---|---|---|---|---|---|---|---|---|
| **FootCapture (HD)** | **3.43** | **2.66** | **0.49** | **1.02** | 1.30 | 1.29 | 1.88 | **1.72**±1.02 | **1.44**±0.75 |
| **GuidedCapture (HD)** | 11.30 | 4.56 | 1.02 | 1.57 | **0.83** | **0.89** | **1.30** | 3.07±3.86 | 1.69±1.43 |
| **FootCapture (NSD)** | **4.58** | **2.65** | **0.49** | **1.05** | 1.30 | 1.30 | 1.93 | **1.90**±1.36 | **1.45**±0.75 |
| **GuidedCapture (NSD)** | 11.04 | 4.56 | 1.02 | 1.56 | **0.90** | **0.91** | **1.38** | 3.05±3.75 | 1.72±1.42 |

Table 1: Mean values in millimeters of the Hausdorff distance (HD) and normalized surface distance (NSD), contrasting FootCapture (Ours) and Guided Capture (Apple). Each model evaluated in the same row was taken of the same foot.

## 4. Discussion and Conclusion

Within this work, we present FootCapture, an application suitable for capturing high-quality images and depth maps, suitable for rendering 3D foot models. FootCapture seems to enable a more consistent and overall superior performance in generating detailed 3D models of feet with roughly equal ease of use. Its ability to deliver precise images for 3D modeling provides the flexibility of separating the process of capturing the images from the process of rendering the models. FootCapture also turned out to be more robust when handling less-than-ideal user input. Consequently, the FootCapture app positions itself as a valuable tool, worth considering for clinicians managing chronic foot wounds.

---

4. https://www.artec3d.com/portable-3d-scanners/artec-leo

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
