# OpenReview forum: "FootCapture: Towards an AR-based System for 3D Foot Object Acquisition through Photogrammetry"
_MIDL.io/2024/Short_Papers — MIDL 2024 Short Papers_

### Official Review · Reviewer_2emK · 2024-04-17

**Confidence:** 5
**Final Rating:** 3.5

**Review:**

The manuscript describes a user study comparing two different user guidance strategies to acquire images for a 3D reconstruction in the podiatrist setting.

While I am generally supportive of these types of studies as I believe that in the MICCAI adjacent communities we do not do enough of this work, my enthusiasm for this specific study is limited. The primary reasons are:
1) It is unclear whether indeed the quantitative results provided in Table 1 isolate the user guidance paradigm. The Apple capture app may use a different number of frames, a different reconstruction algorithm, different post processing (such as smoothing etc.) than the proposed version, and so it's not clear whether we can conclude anything about quantitative performance. This is further exacerbated by the fact that the GT is provided by yet another scanner that will process the data in specific ways.
2) The guidance paradigm here is not shown (it is described as a dome), which is not new. In fact, there is an app doing exactly this on the app store (see [1]).

in conclusion, these limitations (especially limitation 1) question the most important conclusions namely the ones around "foot capture enabling a more consistent and overall superior performance", simply because we might be comparing two quite different things. If the paper is, in fact, about both the app development as well as the user study, then there is insufficient information and detail to evaluate the manuscript in that regard.

Overall, in isolation this paper would fall on the rejection side of the metric, but considering the paper quality overall in my batch of submissions in this paper category I am leaning borderline accept.

[1] https://apps.apple.com/us/app/qlone-3d-scanner/id1229460906?platform=iphone

---

### Decision · Program_Chairs · 2024-04-26

Accept